# Voluntary Language Switching in the Context of Bilingual Aphasia

**DOI:** 10.3390/bs10090141

**Published:** 2020-09-18

**Authors:** Nicholas Grunden, Giorgio Piazza, Carmen García-Sánchez, Marco Calabria

**Affiliations:** 1Center for Brain and Cognition, Pompeu Fabra University, 08005 Barcelona, Spain; ngrunden3@gmail.com; 2Basque Center on Cognition, Brain and Language (BCBL), 20009 Donostia-San Sebastián, Spain; g.piazza@bcbl.eu; 3Hospital de la Santa Creu i Sant Pau, 08041 Barcelona, Spain; CGarciaS@santpau.cat; 4Faculty of Health Sciences, Universitat Oberta de Catalunya, 08018 Barcelona, Spain

**Keywords:** bilingual aphasia, voluntary language switching, bilingual language control, proactive control, reactive control

## Abstract

As studies of bilingual language control (BLC) seek to explore the underpinnings of bilinguals’ abilities to juggle two languages, different types of language switching tasks have been used to uncover switching and mixing effects and thereby reveal what proactive and reactive control mechanisms are involved in language switching. Voluntary language switching tasks, where a bilingual participant can switch freely between their languages while naming, are being utilized more often due to their greater ecological validity compared to cued switching paradigms. Because this type of task had not yet been applied to language switching in bilingual patients, our study sought to explore voluntary switching in bilinguals with aphasia (BWAs) as well as in healthy bilinguals. In Experiment 1, we replicated previously reported results of switch costs and mixing benefits within our own bilingual population of Catalan-Spanish bilinguals. With Experiment 2, we compared both the performances of BWAs as a group and as individuals against control group performance. Results illustrated a complex picture of language control abilities, indicating varying degrees of association and dissociation between factors of BLC. Given the diversity of impairments in BWAs’ language control mechanisms, we highlight the need to examine BLC at the individual level and through the lens of theoretical cognitive control frameworks in order to further parse out how bilinguals regulate their language switching.

## 1. Introduction

Bilinguals have the uncanny ability to manage their languages. Once achieving moderate proficiency, a bilingual has the power to maintain a language throughout a conversation and avoid blurting out unwanted intrusions from their other languages. Then, given the need, many bilinguals can seamlessly “flip the script” and switch in and out of languages to communicate with the people around them. This set of abilities is usually termed as bilingual language control (BLC) and includes a number of cognitive processes [1,2,3]. Outlined in the Adaptive Control Hypothesis (ACH; [4,5]), a comprehensive description of BLC includes at least eight control processes: goal maintenance, conflict monitoring, interference suppression, salient cue detection, selective response inhibition, task engagement and disengagement, and opportunistic planning.

In order to study how bilinguals effectively switch between languages, most studies thus far have employed experimental tasks with cued switching between languages (e.g., [6,7,8,9,10,11,12,13]; for recent reviews see [14,15]). In these tasks, subjects are explicitly shown what language they need to name a given stimulus in with a visual cue (e.g., a color, flag of a given language region, etc.) From the resulting data, it is possible to measure two types of cost, switch and mixing costs. Switch costs are calculated as the difference in naming latencies between “switch” trials, where the target language changed compared to the previous trial, and “repeat” or “stay” trials, where the naming language was the same as in the previous trial. Mixing costs are calculated as the difference between repeat and single trials, where single trials are those named in the non-mixed conditions, such that only one naming language is in use (e.g., [16,17]). However, in light of more recent findings of mixing benefits in some bilingual populations [18,19], it would be more accurate to refer to these general measures as *switching effects* and *mixing effects*. Conceptually, switching effects can be thought of as reflections of the ability to resolve cross-language interference, language engagement, and disengagement, while mixing effects are related to working memory mechanisms, such as the demand in maintaining task goals that are present in a dual-language situation [17,20,21].

Research with pathological populations performing these tasks has reported useful insights on how BLC works. For instance, Calabria et al. [22] administered a cued language switching task to a patient with pathological language switching, with results showing that the patient’s performance on the task was exactly the same as her performance in the more naturalistic connected speech condition. That is, the patient exhibited cross-language intrusions from their non-dominant into their dominant language in the switching task, just as they did when they were required to describe complex pictures or when they engaged in normal conversation. Furthermore, language switching tasks have been useful in assessing the integrity of BLC mechanisms. Calabria et al. [23] found that, in semantic dementia, switching abilities measured via language switching tasks may be spared despite a marked degradation of semantic memory and anomic deficits. Finally, a series of studies in patients with neurodegeneration in the basal ganglia have highlighted an increased impact on BLC deficits compared to other control mechanisms; results indicated that language switching abilities can be more affected than non-linguistic control abilities in bilingual patients with Parkinson’s disease (PD) [20] and that language switching is clearly dissociated from other language control abilities [21]. These studies thus reveal a crucial distinction between BLC deficits affecting control pathways in bilingual language production and generalized language deficits, as PD patients did not exhibit any type of language disorder.

Given these findings, it is plausible then to hypothesize that patients with aphasia may also have some specific impairment in language switching abilities. There is evidence that, in some circumstances, bilingual patients with aphasia may show involuntary cross-language intrusions and/or language mixing by blending morphological features of the two languages within a word or words in a sentence [24,25,26,27,28,29,30,31]. More recently, we have also uncovered evidence that patients with aphasia without involuntary language mixing or switching have shown language control deficits at the lexical level that prevent them from performing effective word retrieval [32,33]. In bilinguals with aphasia (BWA) such as these who do not show pronounced pathological switching, language difficulties have been argued to stem from problems in controlling and managing the inhibition of their languages [34,35]. Therefore, it is reasonable to speculate that their control deficits may extend to difficulties in language disengagement and engagement, as well as language maintenance when they find themselves in contexts where switching back and forth between the two languages is required. However, little is currently known about these deficits in bilingual individuals with post-stroke aphasia that do not demonstrate overt involuntary language mixing or switching.

Addressing this issue, we aimed to identify the BLC deficits that may prevent BWAs from efficiently engaging with their language switching abilities. To do so, we explored the performance of Catalan-Spanish BWA and healthy controls on a language switching task, specifically focusing on two key aspects.

First, we used a voluntary language switching task as a relatively new experimental approach to this issue and a method of capturing a more natural switching behavior for our population of bilinguals. Voluntary switching is regarded as a more ecological measure of language switching [3], especially for bilinguals immersed in dual-language or dense code-switching contexts [4] such as is the case for most Catalan-Spanish bilinguals in the Barcelona metropolitan area. In past studies, this type of language switching task has been employed as an alternative to cued switching paradigms when studying the underlying control mechanisms of endogenous language switching within healthy individuals [18,19,36,37,38] but there is no study to date that incorporates patients with bilingual aphasia. Consequently, any findings in the voluntary switching task for BWA will hold clinical and ecological significance.

Second, our study of language switching abilities in BWAs focused on two main control components measured via switching and mixing effects. In an ongoing debate surrounding the nature of BLC mechanisms, it has recently been proposed that these control mechanisms are differentially involved in single- vs. dual-language contexts of speech production (e.g., [16,17]). Furthermore, recent findings from cued language switching tasks in bilingual patients with PD have demonstrated that switch and mixing costs are possibly related to two qualitatively different mechanisms [20,21]. Within the context of the dual mechanisms of control (DMC) framework of non-linguistic executive control [39,40], these two dual-language effects have been associated with two different types of control, *reactive control* and *proactive control*. Reactive control, measured by switching effects, is defined as a bottom-up, transient, and stimulus-driven type of control whereas proactive control, measured by mixing effects, is top-down, more sustained, and goal-directed. This dual-mechanisms concept of control has subsequently been applied to explain the underlying mechanisms of BLC [17]: reactive control is engaged when bilinguals have to solve for cross-language interference in switch trials, while proactive control comes into play when they have to maintain their two languages active during a dual-language naming condition.

Informing our current study, these two control mechanisms have been studied within the context of voluntary language switching with varying results. In some cases, it has been reported that participants did not show switch costs when utilizing certain voluntary naming strategies, suggesting an effective reliance upon reactive control (“bottom-up switching”; [41]). However, other experiments have shown the opposite, where participants still had switch costs in the voluntary switching condition [18,19]. In contrast to cued language switching, proactive control for bilinguals in voluntary language switching has largely revealed a mixing benefit, where bilinguals named pictures faster in mixed-language vs. single-language contexts [18,19,38].

### Present Study

Our study involved two experiments: Experiment 1, where we sought to replicate previous studies’ exploration of voluntary language switching [18] within our distinct bilingual population of Catalan-Spanish bilinguals; and Experiment 2, where we compared performance of BWAs and healthy bilingual controls on the voluntary language switching task. In both experiments, we tested participants with the following linguistic profile: early (learning both languages before 6 years old) and balanced (largely equivalent frequency of language usage) bilinguals with high levels of proficiency in both languages.

Within Experiment 1, young Catalan-Spanish bilinguals participated in a picture naming task involving blocks with varying target languages: two single-language blocks (one in Spanish and one in Catalan) followed by a dual-language block where they were instructed to name the picture in either language. The design of the experiment is a replication of the de Bruin et al. [18] study, with the exception of cognate status in stimuli; in our study, half of the trials were cognate words (with phonological overlap between the Catalan and Spanish words for a given item) and half were non-cognate words (without phonological overlap). Because the modification is minor, we largely expected similar results to those obtained by de Bruin et al. [18]. First, we expected that the frequency of language switching would be around 50%, where about half of the trials in the dual-language conditions will be named in Catalan and about half in Spanish. Second, as was found by de Bruin et al. [18], we expected a mixing benefit for our bilinguals, measured by larger naming latencies for single trials (single-language condition) compared to repeat trials (dual-language condition). Third, despite this hypothesized benefit of mixing for bilinguals in the dual-language condition, we also expected to find a switch cost within the dual-language context. This would suggest that switching and the usage of language engagement and disengagement control processes would elicit a cost for the language system. The magnitude of the switch cost is expected to be the same for the two languages based on previous studies’ findings with high-proficient bilinguals in cued language switching [9,10,42,43] and by de Bruin et al. [18] in voluntary language switching. Finally, as an established body of literature maintains the presence of a cognate facilitation effect in healthy bilinguals [44,45], we likewise predict that our participants will benefit in cognate trials and demonstrate smaller naming latencies.

Within Experiment 2, Catalan-Spanish bilinguals with aphasia and age-matched healthy controls completed a voluntary language switching task, as in Experiment 1. For the BWA group, we centered our predictions on the hypothesis that BWA may show deficits in monitoring their two languages during a dual-language naming condition. Evidence supporting this hypothesis comes from our previous study with bilingual aphasia, in which we looked at the relationship between semantic interference in picture naming and conflict monitoring [32]. Results showed that bilinguals with aphasia responded slower compared to healthy controls when performing an executive control task requiring them to monitor for both congruent and incongruent stimuli (a flanker task) but were as efficient as healthy controls when they had to solve for stimulus incongruence (conflict cost). Critically, BWA performance on the flanker task correlated with their delayed naming latencies on a picture naming task and was thus interpreted as evidence of a generalized conflict monitoring deficit. In the current experiment, deficits in monitoring could have repercussions when attempting to maintain two languages active in the dual-language condition and impact the proactive control system (for proactive non-linguistic deficits in bilingual patients with aphasia, see [46]).

In light of these previous findings, our predictions for BWA compared to controls were the following: First, if BWAs have deficits that would greatly hinder their performance in the dual-language condition, we predicted that their switching frequency would be lower than 50%, as they presumably stick more to one language rather than alternating between the two. Second, if they do exhibit a lower rate of switching than controls, we predicted that the performance in the dual-language condition for BWAs would resemble the single-language condition and thus we would not expect to find a mixing benefit reported in previous studies (de Bruin et al., 2018). Third, we expected that patients with aphasia and healthy controls would show similar magnitudes of switch costs due to previous evidence that bilinguals with aphasia demonstrate preserved reactive control on a non-linguistic control task [46]. Finally, as has been established in healthy bilinguals as well as bilinguals with aphasia [47], we predicted that BWAs would benefit from a cognate facilitation effect.

## 2. Experiment 1: Young Adult Bilinguals

### 2.1. Methods

#### 2.1.1. Participants

A total of 20 university-age, healthy participants (12 women, 8 men) were recruited from a volunteer database at Pompeu Fabra University. Sociodemographic and language background information was collected via a pre-experimental questionnaire (see Table 1). All participants gave written informed consent in accordance with the Declaration of Helsinki. The experimental protocol was approved by the “Parc de Salut MAR” Research Ethics Committee (reference number: 2018/8029/I). Participants were compensated for their time during the experiment.

In the questionnaire given to participants, age of acquisition and degrees of proficiency in reading, writing, speaking, and comprehension (rated on 7-point scales, with “1” indicating low proficiency and “7” indicating high proficiency) were recorded. Based on responses, subjects who took part in this study were all considered to be early, balanced Catalan-Spanish bilinguals, as indicated by their acquisition of both their languages before 5 years of age and having balanced proficiency in both languages. Additionally, because these measures did not indicate a clear L1 or L2, we labeled the participants’ languages as their dominant language (DL) and non-dominant language (NDL), according to self-reported dominance. In our sample, 10 participants were labeled as Catalan-dominant and the remaining 10 participants were Spanish-dominant. Frequency of usage of both Spanish and Catalan was reported as well. Finally, in order to obtain measures of switching behavior outside the experimental setting, participants’ switching was assessed using the Bilingual Switching Questionnaire (BSWQ; [48]). This questionnaire yielded 5 switching scores: L1-Switch, L2-Switch, Contextual Switching, Unintended Switching, and Overall Switching. Following our classification of our bilinguals’ two languages, L1-Switch and L2-Switch were termed DL-Switch and NDL-Switch, respectively. See Table 1 for sociodemographic and language information.

#### 2.1.2. Procedure

The visual stimuli for this task consisted of 60 different pictures taken from Snodgrass and Vanderwart [49]. Stimuli were balanced across Catalan and Spanish for measures of logarithmic frequency (Catalan: *M* = 1.19, *SD* = 0.52; Spanish: *M* = 1.11, *SD* = 0.46; *t*(59) = 0.90, *p* = 0.39), word length (Catalan: *M* = 5.86, *SD* = 1.44; Spanish: *M* = 5.70, *SD* = 1.25; *t*(59) = 0.66, *p* = 0.41), and number of phonemes (Catalan: *M* = 5.27, *SD* = 1.32; Spanish: *M* = 5.35, *SD* = 1.25; *t*(59) = 0.82, *p* = 0.91). The frequencies for the Spanish and Catalan names were obtained from the LEXESP [50] and the Catalan Dictionary of Frequencies [51] databases, respectively. In addition to these linguistic factors, half of the picture names were cognate words while the other half were non-cognate words between the two languages.

At the onset of the experiment, subjects underwent a familiarization exercise where they were presented with all experimental pictures and asked to read aloud the correct responses for each picture printed below it in both Spanish and Catalan. While stimuli with high name agreement were selected for this experiment, this initial presentation of pictures served to strengthen this name agreement of stimuli across participants.

Experimental tasks included three blocks: two single-language blocks and a dual-language block. Each single-language block consisted of 120 naming trials (60 pictures presented 2 times each) in either Spanish or Catalan. The order of these single-language blocks was counterbalanced across subjects and instructions for the task were always given in the target language of the block. These single-language blocks were then followed by a dual-language block (360 trials, 60 pictures repeated 6 times each), where participants were asked to name items with “whichever language comes most naturally” but also to “switch languages multiple times” throughout the task. The language of these instructions was counterbalanced across participants so as to balance any priming effects in subsequent naming. Dual-language blocks included two catch trials at the beginning of the picture naming which were then disregarded during analysis. Because of the longer duration of these blocks, subjects were given two opportunities to rest during the naming trials. Each of these breaks was also followed by another catch trial that was then taken out for analyses.

For all trials regardless of block, subjects were presented with a fixation cross for 500 ms, followed by the image to be named by the participant. Said image remained on the screen for 2000 ms (irrespective of when a participant began/ended their response) and audio was recorded during this time to capture responses given. The picture naming task was administered to participants using DMDX software [52].

#### 2.1.3. Data Analysis

Out of the 20 young bilingual participants recruited for this study, 1 participant was excluded due to an abnormally high number of errors (42 errors; group error average = 11.95). The remaining 19 participants were included in all subsequent statistical analyses.

The dependent variables for this study, naming latencies and accuracy of participants’ responses, were analyzed off-line with CheckVocal [53]. For each audio file, the beginning of the participant’s response was marked and the response was coded as correct or incorrect. Naming latencies for incorrect responses and those exceeding 2 SDs above or below a given subject’s mean naming latency were excluded. Naming errors were classified under the following types [54]: (a) omissions: non-intelligible verbal response given during recording window or no response; (b) semantic errors: verbal response semantically related to target word; (c) formal errors: deletion, substitution, or addition of phonemes in target word; (d) unrelated errors: verbal response with no semantic or other relation to target word; (e) cross-language intrusions: correct naming of picture but in non-target language (only applicable for single-language blocks); and (f) auto-correction: incorrect verbal response followed by correction with target word.

In order to examine effects of switching and mixing, trials in single-language blocks were classified as “single” trials whereas, in dual-language blocks, trials were labeled as “repeat” or “switch”, depending on whether the language used to name an item was the same (repeat) or different (switch) compared to the previous trial. For error trials that were able to be categorized as either Spanish or Catalan (semantic errors, non-ambiguous formal errors, unrelated errors and cross-language intrusions), the following trial was classified as repeat or switch trials in reference to the language of the error. However, for error trials that were not able to be categorized by language (i.e., no response and ambiguous formal errors), the trials directly after them were unclassifiable as repeat/switch and were excluded from analyses. Overall, an average of 4.58 trials (SD = 6.27), with a maximum of 25 trials, were unclassifiable per subject.

With this data, we first analyzed the distribution of the switch trials per language and per cognate status in order to assess whether the switching frequency was around 50%, as done by de Bruin et al. [18]. Subsequently, we ran two separate repeated-measures ANOVAs for RTs and accuracy as dependent variables, including Trial Type (Single, Repeat, or Switch), Language (Dominant vs. Non-dominant) and Cognate Status (Cognate vs. Non-cognate) as within-subject factors. If the assumption of sphericity (Mauchly’s test) was violated, the Greenhouse–Geisser correction was applied by adjusting the degrees of freedom.

### 2.2. Results

**Switching frequencies.** Participants on average switched on 43.99% (*SD* = 7.43) of the mixed language trials. Of those switch trials, there were similar percentages of switching for cognates (*M* = 49.73, *SD* = 3.26) and non-cognates (*M* = 50.27, *SD* = 3.26; *t*(18) = −0.37, *p* = 0.718). Likewise, switch trials were equally shared between switches into dominant language (*M* = 50.31, *SD* = 0.97) and switches into non-dominant language (*M* = 49.69, *SD* = 0.97; *t*(18) = 1.39, *p* = 0.182).

**Naming latencies.** Analyses showed that the main effect of Trial Type, *F*(1.17, 36) = 5.52, *p* = 0.02, η_p_^2^ = 0.24, was significant. Post hoc analyses revealed a significant switch cost for participants, where switch trials (M = 769 ms, SD = 96) were significantly slower than repeat trials (M = 753 ms, SD = 91; *p* = 0.002). The difference in magnitude of switch cost was not significantly different across languages (DL switch cost: M = 2.36 ms, SD = 3.00; NDL switch cost: M = 2.07, SD = 3.79; *t*(18) = 0.245, *p* = 0.809). Furthermore, naming for single trials (M = 788 ms, SD = 95) was significantly slower than for repeat trials (M = 753 ms, SD = 91; *p* = 0.04), suggesting a mixing benefit, where participants improved their naming latencies in dual-language conditions compared to single-language conditions (see Figure 1).

The main effect of Cognate Status was also significant, *F*(1,18) = 25.77, *p* < 0.001, η_p_^2^ = 0.59, where cognates (*M* = 759 ms, *SD* = 88) were named faster than non-cognates (*M* = 783 ms, *SD* = 93). Neither the main effect of Language, *F*(1,18) = 0.18, *p* = 0.68, nor any interactions between factors were found to be significant.

**Accuracy.** Analyses revealed a significant main effect in Trial Type, *F*(1.51, 36) = 25.14, *p* < 0.001, η_p_^2^ = 0.58. Post hoc tests for Trial Type effects showed a significant decrease in accuracy during single trials (*M* = 97.24%, *SD* = 1.6) compared to both repeat (*M* = 99.12%, *SD* = 0.61; *p* < 0.001) and switch (*M* = 99.17%, *SD* = 0.78; *p* < 0.001) trials. The main effect of Cognate Status was also significant, *F*(1,18) = 4.87, *p* = 0.04, η_p_^2^ = 0.21, where cognates (*M* = 98.51%, *SD* = 1.43) were named more accurately than non-cognates (*M* = 97.50%, *SD* = 1.47). The main effect of Language, *F*(1,18) = 2.99, *p* = 0.101, was not found to be significant.

The Cognate Status × Trial Type interaction was also significant, *F*(2, 36) = 3.92, *p* = 0.03, η_p_^2^ = 0.18. Post hoc analyses showed that non-cognate single trials (*M* = 96.24%, *SD* = 2.86) were named significantly less accurately than all other non-cognate trials and all cognate trials. There was also a significant difference in accuracy between cognate single trials (*M* = 97.94%, *SD* = 2.76) and cognate repeat trials (*M* = 99.47%, *SD* = 1.05).

**Correlations between Language Measures and Voluntary Switching Effects.** To investigate whether participants’ language backgrounds affected task performance, we ran correlations between language measures and switching/mixing effects. Correlation analyses were run comparing the magnitudes of switching and mixing effects (calculated as an individual’s switch cost divided by the group averages for repeat and switch trials × 100) to all BSWQ scores, ages of acquisition, proficiency scores, and frequency of usage of both dominant and non-dominant languages. No significant correlations were found between measures of language profile and switching or mixing effects.

## 3. Experiment 2: Bilinguals with Aphasia and Healthy Controls

### 3.1. Methods

#### 3.1.1. Participants

Seven Catalan-Spanish bilinguals with aphasia (BWA) were recruited to take part in this study from the Speech Pathology Clinic at the Hospital de la Santa Creu i Sant Pau. Ten age-matched bilingual adults were also included in this experiment as neurologically healthy controls. Prior to the experiment, all participants completed a language background questionnaire where they were asked to report sociodemographic and linguistic factors, as in Experiment 1. In this experiment, language usage was measured with a set of questions about the frequency of Spanish and Catalan usage in different settings and across different periods of the individual’s life. These responses were transformed into a final score, expressed as a percentage of usage where 0% was solely Spanish, 100% was solely Catalan, and 50% marked a balanced usage of the two languages. Furthermore, all responses on the language questionnaires, with the exception of those in the BSWQ, were given by BWAs in reference to premorbid levels. See Table 2 for comparisons on sociodemographic and linguistic characteristics between groups.

This study was carried out in accordance with the recommendations of the “Parc de Salut MAR-Research Ethics Committee.” All participants gave written informed consent in accordance with the Declaration of Helsinki and with the research protocol that was approved by the “Parc de Salut MAR-Research Ethics Committee” (reference number: 2018/8029/I).

#### 3.1.2. Language Assessment

Along with sociodemographic and language background data, clinical data for BWAs were also compiled. Patients were assessed using the Spanish version of the Western Aphasia Battery (WAB; [55]) in order to ascertain their current type and severity of aphasia. This clinical assessment tool was administered and scored by a clinical neuropsychologist at Hospital de la Santa Creu i Sant Pau. The WAB is a comprehensive test of language functions with a relatively short test administration time (30–60 min) and includes four language subtests which assess spontaneous speech, comprehension, repetition, and naming to calculate an Aphasia Quotient (AQ). Based on this score, the severity and type of aphasia can be determined. In our sample, 4 BWAs were classified as having mild aphasia and the remaining 3 as having moderate aphasia. In terms of aphasia type, 3 had anomic aphasia, 2 had Wernicke’s aphasia, 1 with conduction aphasia, and 1 with transcortical motor aphasia. Patients were only tested in Spanish as a Catalan version of the WAB is not currently available. The etiologies and months since the onset of brain lesions were also reported for each patient. Finally, patients did not present any clinically significant motor speech disorders at the time of the experiment. See Table 3 for clinical data.

Patients’ language abilities were further tested using part C of the Bilingual Aphasia Test (BAT; [56]), which assesses cross-language abilities over four subtests: Word Recognition, Word Translation, Sentence Translation, and Grammatical Judgment. In Word Recognition, patients were asked to select the correct translation for each word from a list of 10 possible choices (5 words per language; max. score = 10). In the Word Translation task, patients needed to verbally supply the translation of a word spoken by the examiner (10 words per language; max. score = 20). Increasing in difficulty, subjects then were asked in the Sentence Translation task to provide a translation of a sentence that could be repeated a maximum of three times by the examiner (scoring based on correct translations of 3 sections of each sentence for 6 sentences in each language; max. score = 36). Finally, in Grammatical Judgment, patients were asked to determine whether a sentence spoken by the examiner was grammatically correct and, if incorrect, how to fix it (scoring based on correct judgment of grammatical structure and accurate correction of grammatical mistakes if applicable for 8 sentences per language; max. score = 28). These subtests of the BAT-C were administered by a bilingual neuropsychologist, completing all four tasks in one direction of translation followed by the same four tasks in the other direction (i.e., Catalan to Spanish in all tasks followed by Spanish to Catalan, or vice versa).

#### 3.1.3. Procedure

The experimental procedure is the same as Experiment 1, with the exception of the number of trials for patients being adapted to avoid fatigue. While maintaining the same ratios of cognates, the quantity of trials and pictures were halved: patients were presented with 60 naming trials (30 pictures presented 2 times each) for single-language blocks and 180 trials (30 pictures repeated 6 times each) for dual-language blocks. Stimuli were balanced across Catalan and Spanish for measures of frequency (Catalan: *M* = 1.29, *SD* = 0.56; Spanish: *M* = 1.20, *SD* = 0.51; *t*(29) = -1.25, *p* = 0.22), word length (Catalan: *M* = 6.03, *SD* = 1.61; Spanish: *M* = 6.10, *SD* = 1.29; *t*(29) = 0.23, *p* = 0.82), and number of phonemes (Catalan: *M* = 5.57, *SD* = 1.46; Spanish: *M* = 5.80, *SD* = 1.38; *t*(29) = 0.88, *p* = 0.39). The frequencies for the Spanish and Catalan names were obtained from the LEXESP [50] and the Catalan Dictionary of Frequencies [51] databases, respectively. Additionally, in each trial, picture presentation and audio recording were extended to 3500 ms in anticipation of slower picture naming for BWA.

#### 3.1.4. Data Analysis

Sociodemographic and linguistic variables were compared between the two groups using independent-samples t-tests or one-sample t-tests when no variability was present in one of the two groups (i.e., for proficiency measures rated at maximum).

For experimental data, dependent variables (naming latencies and accuracy of responses) for this experiment mirror those of Experiment 1. Naming error coding included the same categories as Experiment 1, with a key exception. As BWA were expected to commit more errors unable to be categorized as either switch or repeat trials, trial classification was extended to include language information of answers uttered outside of the 3500 ms response window. While these trials were still considered omission errors, the language of these late responses permitted the following trials to be identified as switch or repeat and thus provided more accurate representation of switching behavior in our participants. With this alteration, an average of 6.60 trials (*SD* = 6.19) with a maximum of 16 trials were excluded for the control group and an average of 12.57 trials (*SD* = 6.40) with a maximum of 25 trials were excluded for the BWA group.

First, we analyzed the distribution of the switch and repeat trials per language and per cognate status in order to assess group differences for the switching frequency. This measure (calculated as number of switch trials over the total number of classifiable dual-language block trials) was compared between BWA and controls with both a parametric (independent samples t-test) and a non-parametric test (Mann–Whitney U test).

Subsequently, we ran two separate repeated-measures ANOVAs for RTs and accuracy as dependent variables, including Trial Type (Single, Repeat, or Switch), Language (Dominant vs. Non-dominant) and Cognate Status (Cognate vs. Non-cognate) as within-subject factors, and Group (Patients vs. Healthy controls) as a between-subject factor. Follow-up analyses were conducted for significant three- and four-way interactions. If the assumption of sphericity (Mauchly’s test) was violated, the Greenhouse–Geisser correction was applied by adjusting the degrees of freedom.

Finally, we performed independent sample t-tests to compare word durations for BWA and control groups.

### 3.2. Results

**Sociodemographic and linguistic variables.** There were no significant differences in sociodemographic measures or premorbid linguistic factors (age of acquisition, proficiency, or language usage) between the two groups. For (post-morbid) language switching behavior, BWAs reported higher degrees of switching into their dominant (BWAs: *M* = 9.86, *SD* = 1.86; controls: *M* = 7.80, *SD* = 1.14; *p* = 0.012) and non-dominant languages (BWAs: *M* = 9.57, *SD* = 0.79; controls: *M* = 6.30, *SD* = 1.57; *p* < 0.001) compared to controls (see Table 2). For each participant, we compared the scores of the BAT-C of the two languages using a Chi-squared test with Yates’ correction; five out of seven patients showed parallel language deficits (Pt 2 showed a significantly more impaired score in their non-dominant compared to their dominant language, and Pt 3 the opposite).

**Switching frequencies.** Both experimental groups switched between their languages in approximately half the trials (BWA: *M* = 49.65%, *SD* = 13.15; Controls: *M* = 47.97%, *SD* = 7.32), with no significant differences between groups, *t*(15) = 0.34, *p* = 0.739; U = 40.00, *p* = 0.669. Additionally, participants did not show any differences in switching associated with cognate status [cognates: *M* = 50.06%, *SD* = 3.43; non-cognates: *M* = 49.94%, *SD* = 3.43; *t*(16) = 0.07, *p* = 0.943] or language dominance [DL: *M* = 50.01%, *SD* = 1.07; NDL: *M* = 49.99%, *SD* = 1.07; *t*(16) = 0.01, *p* = 0.99].

**Naming latencies.** Analyses revealed a main effect of Trial Type, *F*(2, 30) = 3.26, *p* = 0.05, η_p_^2^ = 0.18. Post hoc analyses showed that participants demonstrated significant switch costs, as they were slower in naming switch trials (*M* = 1176 ms, *SD* = 55) than naming repeat trials (*M* = 1072 ms, *SD* = 35, *p* < 0.05). However, they did not show significant mixing costs, as naming latencies between repeat trials (*M* = 1113 ms, *SD* = 38) and single trials (*M* = 1072 ms, *SD* = 35) were not significantly different (*p* = 0.31).

The main effect of Cognate Status was also significant, *F*(1,15) = 30.09, *p* < 0.001, η_p_^2^ = 0.67, where cognate trials (*M* = 1089 ms, *SD* = 34) were named faster than non-cognate trials (*M* = 1151 ms, *SD* = 40). Conversely, the main effect of Language was not significant, *F*(1,15) = 3.01, *p* = 0.10. Finally, the main effect of Group was significant, *F*(1,15) = 44.97, *p* < 0.001, η_p_^2^ = 0.75, with greater naming latencies for BWA (*M* = 1363 ms, *SD* = 55) than for controls (*M* = 877 ms, *SD* = 46). See Figure 2 for comparison between groups.

The interaction between Cognate Status and Group was also significant, *F*(1,15) = 15.21, *p* = 0.001, η_p_^2^ = 0.50. An independent samples t-test was performed on the difference in magnitude of the cognate effects between the two groups and we found that this effect was larger in patients (*M* = 107 ms, *SD* = 69) than in controls (*M* = 18 ms, *SD* = 18; *t*(16) = 3.01, *p* = 0.001).

Furthermore, both the Language × Cognate Status × Trial Type interaction, *F*(2,30) = 6.66, *p* < 0.01, η_p_^2^ = 0.31, as well as the Language × Cognate Status × Trial Type × Group interaction, *F*(2,30) = 3.54, *p* < 0.05, η_p_^2^ = 0.19, were significant. To address these complex interactions, follow-up analyses were conducted by performing repeated-measures ANOVAs for cognates and non-cognates separately, both including Language and Trial Type as within subject factors. Said analyses were also separated between control and patient groups.

In controls, the main effect of Trial Type was significant for both cognates, *F*(2,18) = 3.46, *p* = 0.05, η_p_^2^ = 0.44, and non-cognates, *F*(2,18) = 5.52, *p* = 0.01, η_p_^2^ = 0.38, in the dominant language; and for cognates, *F*(2,18) = 7.89, *p* = 0.003, η_p_^2^ = 0.47, but not for non-cognates, *F*(2,18) = 3.25, *p* = 0.06, in the non-dominant language. Significant switch costs were found for cognates (repeat trials: *M* = 850 ms, *SD* = 72; switch trials: *M* = 900 ms, *SD* = 91; *p* = 0.03) and non-cognates (repeat trials: *M* = 870 ms, *SD* = 54; switch trials: *M* = 917ms, *SD* = 82; *p* = 0.02) in the dominant language and only for cognates in the non-dominant language (repeat trials: *M* = 877 ms, *SD* = 56; switch trials: *M* = 917 ms, *SD* = 96; *p* = 0.04). Mixing costs were only significant for cognates in the non-dominant language (single trials: *M* = 811 ms, *SD* = 75; repeat trials: *M* = 877 ms, *SD* = 56; *p* = 0.03).

In patients, the main effect of Trial Type was neither significant in the dominant [cognates words: *F*(2,12) = 0.11, *p* = 0.89; non-cognates words: *F*(2,12) = 1.11, *p* = 0.36] nor in the non-dominant language [cognates words: *F*(2,12) = 3.41, *p* = 0.07; non-cognates words: *F*(2,12) = 1.17, *p* = 0.34].

**Individual level analyses for naming latencies.** As we observed a great variability in the patient group for switching and mixing effects, we ran individual level analyses for BWAs. We first calculated individual proportional switching and mixing effects and we used a modified t-test described by Crawford and Howell (1998) for independent samples to compare each individual’s performance to the controls’ mean (switch cost: *M* = 3.2, *SD* = 3.4; mixing cost: *M* = 2.2, *SD* = 3.3). The t values were calculated as follows:$$t=\frac{X_{1}-X_{2}}{s^{2}\;\sqrt{\frac{X_{2}+1}{N}}}$$
where *X*_1_ is the individual’s performance, *X*_2_ is the mean of the control sample, *s*^2^ is the standard deviation of the control group, and *N* is the sample size.

The results of the analyses showed that 2 patients (Pt 2: 24.5%, *p* < 0.001; Pt 5: 13.3%, *p* < 0.01) had larger switch costs as compared to controls. Moreover, 3 patients (Pt 2: 25.6%; Pt 6: 15.2%, Pt 7: 15.7%; all *p*-values < 0.001) had larger mixing costs than controls and one (Pt 4: −31.2%, *p* < 0.001) had a significant mixing benefit as compared to controls (see Figure 3).

**Accuracy.** The main effect of Trial Type was significant, *F*(2,30) = 19.01, *p* < 0.001, η_p_^2^ = 0.56, and *post hoc* analyses revealed a significant decrease in accuracy for single trials (*M* = 86.65%, *SD* = 10.67) compared to both repeat (*M* = 94.27%, *SD* = 4.68; *p* < 0.001) and switch trials (*M* = 94.38%, *SD* = 6.37; *p* < 0.001) trials. The main effects of Language, *F* (1,15) = 0.37, *p* = 0.56, and Cognate Status, *F*(1,15) = 2.27, *p* = 0.18, were not significant. Finally, the main effect for Group was significant, *F*(1,15) = 25.91, *p* < 0.001, η_p_^2^ = 0.62, where control responses (*M* = 97.21%, *SD* = 2.58) were more accurate than BWA responses (*M* = 86.47%, *SD* = 6.42; *p* < 0.001).

Furthermore, the Cognate Status × Group interaction was significant, *F*(1,15) = 5.86, *p* = 0.03, η_p_^2^ = 0.28. *Post hoc* analyses showed that controls performed with the same accuracy for cognates (96.20%) and non-cognates (97.40%, *p* = 0.37), whereas patients were more accurate in naming cognates (88.25%) than non-cognates (84.70%, *p* = 0.04).

The Trial Type × Group interaction was also significant, *F*(2,30) = 3.74, *p* = 0.04, η_p_^2^ = 0.20. Post hoc analyses showed that in the BWA group, single trial accuracy (*M* = 78.93%, *SD* = 11.22) was significantly lower than both repeat (*M* = 90.58%, *SD* = 3.50; *p* < 0.05) and switch (*M* = 90.14%, *SD* = 7.52, *p* < 0.05) trials. Similarly, single trial accuracy in controls (*M* = 94.04%, *SD* = 4.67) was significantly lower than both repeat (*M* = 98.42%, *SD* = 2.50; *p* < 0.01) and switch (*M* = 98.43%, *SD* = 2.55, *p* < 0.01) trials. However, the difference in accuracy between single and repeat trials was significantly larger in patients (11.51%) than in controls (4.38%, *p* = 0.05).

**Error distribution.** To parse the distribution of error types for experimental groups within single-language blocks, the number of incorrect trials for each error type was expressed as a percentage of total single-language trials for each participant and then averaged within groups (Table 4). Although there is a clear increase in errors on single-language trials overall, BWAs show a specific rise in the proportion of cross-language intrusions, accounting for an average of ~5% of all trials compared to 0.67% for controls. Of note, 11 out of the 12 auto-correction errors committed by BWAs (accounting for an average of 1.43% of single trials) were errors where they first produced the picture name in the incorrect language and then corrected with the target language word, or a “cross-language auto-correction” hybrid. Consequently, cross-language intrusions and these auto-corrections were grouped together as cross-language errors for BWAs, totaling to an average of 6.79% of single-language trials. Of note, these cross-language errors were distributed equally across languages: 52% were NDL intrusions in DL trials and 48% were DL intrusions in NDL trials.

Broken down to individual patient performance on single-language trials (Table 5), cross-language intrusions and auto-corrections combined were the most common errors for 3 BWAs (Pt 1, Pt 5, Pt 6), the second most common errors behind omissions for 2 BWAs (Pt 3 and Pt 7), and were not as common among the remaining 2 BWAs (Pt 2 and Pt 4).

**Word duration.** The main effect of Group was found to be significant, *F*(1,15) = 15.44, *p* = 0.001, η_p_^2^ = 0.51, where the BWA group produced larger word durations (*M* = 730 ms, *SD* = 174) than controls (*M* = 479 ms, *SD* = 86). No other main effects were significant. The 3-way interaction between Language × Cognate Status × Trial Type, *F*(2,30) = 4.18, *p* = 0.03, η_p_^2^ = 0.22, and the 4-way interaction of Language × Cognate Status × Trial Type × Group, *F*(2,30) = 3.96, *p* = 0.03, η_p_^2^ = 0.21, were both significant, but no comparisons remained significant in post hoc analyses.

## 4. Discussion

With two experiments, we aimed to investigate BLC and underlying processes of reactive and proactive control within the context of voluntary language switching in bilinguals. In Experiment 1, we explored this issue in healthy individuals with the main aim to establish a reliable paradigm based on the Bruin et al. [18] study within our own bilingual population. In Experiment 2, we investigated how aphasia may impact these two control mechanisms. We found several results that we discuss below in reference to previous findings from voluntary language switching studies, as well as within the context of bilingual language control models at both group and individual patient levels.

### 4.1. Group Level Analyses

In Experiment 1, results largely mirrored those of the de Bruin et al. [18] study and also supported our initial predictions on switching and mixing effects for our sample of bilinguals. First, as predicted, participants switched and repeated languages trial-by-trial in the dual-language condition at a balanced rate, with switching frequencies remaining around 50% for both languages and for cognates versus non-cognates. This balanced amount of datapoints for switch and repeat trials facilitated subsequent comparisons between these trial types. Second, participant performance did reveal a switch cost, where trials were named slower on average when participants switched languages versus when they used the same language as the previous trial. This cost is similar to that found in previous voluntary language switching studies [18,19,36]. Furthermore, participants showed no difference in magnitude of switch costs between their dominant and non-dominant languages, supporting a lack of language-dependent effects in our highly proficient and balanced bilingual population; this coincides with previous findings in the cued language switching studies with Catalan-Spanish bilinguals [42,43]. We also found a mixing benefit, where naming responses in the dual-language repeat trials were faster and named with better accuracy than in single-language trials. Again, this is in line with de Bruin and colleagues’ findings in balanced Basque-Spanish bilinguals. Although we did not find evidence of a pervasive “bail-out” strategy where bilinguals constantly switched into one language to compensate for the other [36], it is possible that this strategy was used periodically in dual-language blocks when balanced bilinguals struggled to retrieve one language and instead resorted to the other; this would not have been an option in single-language blocks with only one possible language target and would thus yield more errors. Due to this aspect of experimental design that can be capitalized upon by opportunistic switching, as it is understandable for bilinguals to generally demonstrate an accuracy advantage in dual-naming conditions.

In addition to replicating the de Bruin et al. findings, we sought to further explore voluntary switching with both cognate and non-cognate words. As in previous studies of Catalan-Spanish bilinguals [44] as well as studies involving other bilingual populations [45], our bilingual participants in Experiment 1 showed an overall cognate facilitation effect in reduced naming latencies and increased accuracy for cognates compared to non-cognates. This cognate effect did not interact with switching and mixing costs in naming latencies, but cognates did influence accuracy across trial types, where non-cognate, single-language trials were named with the worst accuracy. This interaction seems to indicate that the cognate effect is more potent when both languages are held in a state of generalized activation (dual-language condition) and how cognate phonological similarity is more readily accessible in this condition versus the single-language condition, where only one language is the target of activation. Our study thus adds to the literature of voluntary switching by showing that cognate words could provide greater facilitation of correct responses when two languages are globally engaged and voluntarily toggled between by a bilingual.

Viewed in the context of the Adaptive Control Hypothesis (ACH; [4]) and the dual-mechanisms of control (DMC) framework [39,40], the mixing benefit seen here in Experiment 1 could reflect a highly trained proactive control system within BLC, where young bilinguals who interact in dual-language environments are able to maintain their languages “at the ready” with little difficulty. Furthermore, this mixing benefit indicates that the global activation of two languages has been mastered to the point where it is more costly to have to selectively inhibit one language to perform in a single-language context. Finally, as the switch costs in Experiment 1 mirror those seen in similar voluntary switching studies [18,19,38] and those in cued-switching studies, engaging reactive control mechanisms seems to lead to temporal costs for the average bilingual, unless experimental manipulations or individual strategies are used to specifically favor bottom-up switching [37].

In Experiment 2, we sought to apply the voluntary language switching paradigm to the study of bilinguals with aphasia (BWAs) and how their volitional switching behavior compared with that of age-matched controls. Due to a high degree of variability in this patient population on language performance measures and a modest sample size for both experimental groups, our analyses included group-level comparisons as well as individual-level analyses for BWAs. With these two layers of interpretation, we discuss areas of commonality across groups as well as the heterogeneity of individual performances in our patient sample.

At the group level, BWAs were able to switch with the same frequency between their languages as controls during the dual-language blocks, a finding contrary to our predictions at the onset of this experiment. Initially, we had predicted that patients with aphasia would perform fewer switches than controls when given the option to choose their language, based on previous findings showing the presence of control deficits in BWAs [57,58], especially for conflict monitoring [32]. Although there are cases of pathological and uncontrolled switching in patients with brain lesions (e.g., [22,24,25,27,31,59,60]), our BWA group demonstrates that not all BWA patients are impaired in their ability to voluntarily switch between their languages. One possible explanation for this is that the patients included in this study were all considered to have mild to moderate aphasia and did not report significantly higher rates of unintended or involuntary switching behavior on the BSWQ compared to controls. Moreover, some of the previous studies that have documented pathological language mixing and switching in patients have suggested that this behavior is mainly due to damaged subcortical areas (left caudate, [24]; [31]) or within the fronto-striatal system, as the BLC model would predict for language activation and selection [1]. Therefore, this first result could indicate that our BWAs have spared functionality in these brain areas or at least did not sustain damage to the point of affecting their language switching at a clinical level; however, this is purely speculative given that we did not have access to neuroimaging data detailing patients’ specific lesion locations. To conclude whether BWAs had completely spared BLC, we explored whether they also exhibited spared reactive and proactive control as a group below and how they behaved at an individual level in the next section.

In general, BWAs were overall slower and less accurate in naming than controls, a result that we have previously observed in this clinical population on naming tasks [32,61]. Additionally, the response durations given by BWAs were significantly larger than those of control, but this factor did not interact with other experimental variables; this finding, along with to the lack of clinically significant motor speech disorders, allows us to subsequently discard articulatory processes as factoring into language switching performance for this BWA sample. Furthermore, as in Experiment 1, cognate status was examined here as a factor in switching behavior. Interestingly, there was a greater cognate facilitation in both naming latencies and accuracy for BWAs compared to controls. Adding to other studies which have also found this cognate effect for bilinguals with aphasia [47,62], this finding is particularly relevant for proposed cognate-based interventions in the clinical realm [63], as cognate effects found in a voluntary switching task carry more ecological validity in day-to-day naming for bilinguals and thus indicate more potential for generalizability in cognate benefits.

For reactive control, we did not find a significant difference between the two groups. Indeed, both patients and controls collectively experienced switch costs, where switch trials were named slower than repeat trials. Separated by groups, controls suffered clear switch costs across all levels of cognate status and language, with the exception of non-cognates in their non-dominant language. This impact of switching, together with the similar switching results from Experiment 1, suggests a pervasive cost of switching languages in this task [18,19], where disengaging and engaging with another language requires activation of reactive control processes and, barring specific strategies to circumvent costs, leads to slower naming responses for bilinguals.

For proactive control, the only significant effect of naming in a dual-language context vs. single-language context was a mixing *cost*, where controls named cognates in their non-dominant language slower during repeat trials; all other mixing effects across languages, trial types, and cognate statuses for controls were non-significant. BWAs as a group did not show any significant mixing effects. Thus, our results seem to indicate that global effects of intra-block language mixing are neither beneficial nor costly for balanced adult bilinguals. Mixing effects and proactive control are further examined on the individual level for patients in the following section.

Accuracy on the language switching task for Experiment 2 showed a similar pattern as seen in Experiment 1, where single language trials were named less accurately than repeat trials in the dual-language naming condition; however, this difference was more pronounced in the BWA group. While this pattern is again likely a product of participants having the option to use either language as a “lifeline” in the dual-language condition but not in the single-language condition, BWAs’ larger decrease in single-language accuracy might also reflect deficits in retrieving target language words. Interestingly, if we look at the error distribution for both groups in the single-language blocks, BWAs showed a marked increase in cross-language intrusions, the most frequent type of error for this group (an average of 5.36% of trials for BWAs and 0.67% of trials for controls in single-language blocks; see Table 4) behind omission errors. Furthermore, combining cross-language intrusions with cross-language auto-corrections (see Error Distribution section for definition) under the umbrella term ‘cross-language errors,’ the increased frequency of these errors in BWAs (5.76% of all single-language trials) could be driving the greater difference across trial types in group accuracy.

While the aim of this study was not to compare voluntary language switching performance across different bilingual age groups, results from young adult bilinguals in Experiment 1 and the control group from Experiment 2 did differ in some important respects. Considering the effect of language mixing in dual-language repeat trials versus single-language trials, young bilinguals demonstrated a mixing benefit while older bilinguals did not. This difference contrasts with the findings of de Bruin, Samuel, and Duñabeitia [19], where older bilingual adults, like their younger counterparts, exhibited a mixing benefit. Additionally, while our young bilinguals showed a positive effect of cognates on naming accuracy, we were unable to find this effect in the older bilinguals. Both of these discrepancies in performance deserve to be explored in future studies but, as our focus here remains on results stemming from patients with aphasia, this analysis falls outside the scope of this study.

### 4.2. Individual Level Analyses of Patients

Zooming in on the individual level, the complexities of language switching in patients become more salient. In our sample, individual BWAs showed at least two types of naming performance in their accuracy: some BWAs suffered from cross-language errors in the single naming condition in a much more substantial way (Pt 1 and 5) while most of them did not produce such errors or produced very few (Pt 2, Pt 3, Pt 4, Pt 6, and Pt 7). In light of this, we argue that interpretations of results hold more validity if we take into account these two profiles separately, as they are indicative of different deficits in the control mechanisms underlying language switching.

Within error type distributions, the presence of more cross-language errors in the single naming condition is potentially an indication of some BLC deficits. Reminiscent of how patients with pathological language switching (e.g., [22]) perform but to a lesser degree, certain BWAs were unable to restrict the lexicalization to that specific language during the task and suffered intrusions (whether auto-corrected or not) from the unintended language. However, these patients did not have difficulties in proactive control as they were able to maintain the two languages with no mixing effects when they named items in the dual-language situation. Looking at the data for these two patients, Pt 1 and Pt 5, we can see that they both have mixing performance within the normal range as compared to healthy controls. With this type of performance, we could speculate that these patients had some specific deficits within goal maintenance or interference control, elements of BLC proposed by Green and Abutalebi [4] in the ACH; their cross-language errors suggest difficulties in maintaining the goal of speaking in just one language or of suppressing the cross-language interference from the unintended language. However, deficits for the other control mechanisms (salient cue detection, selective response inhibition, task disengagement, task engagement, and opportunistic planning) proposed by this framework cannot explain the behavior of these patients, as these deficits would impact the dual-language naming conditions as well.

Alternatively, we might also argue that the presence of cross-language errors in the context of aphasia denotes a strategy of compensating for word retrieval deficits in a target language [64] which we are only able to detect when production is restricted to said language [22]; adopting this technique could have clear benefits in dual-language contexts but generalization to single-language contexts would lead to the observed levels of cross-language errors. Evidence for this interpretation comes from significantly higher ratings of DL and NDL switching for patients compared to controls on the BSWQ. The questions comprising these two subscales address switching into the other language when the target language is inaccessible (i.e., “When I cannot recall a word in Catalan, I tend to immediately produce it in Spanish”; [48]) and thus higher ratings would indicate more instances of compensatory language switching when the target language is inaccessible. This type of strategy of cross-language facilitation or cueing with the non-target language has been proposed as a potentially voluntary or involuntary tactic used by BWAs [65]. However, it is important to highlight that this compensation is not necessarily free of switch costs. Indeed, Pt 5 had significantly larger switching costs as compared to controls, suggesting that, despite the patient’s ability to compensate with the other language for some measures, their reactive control was not so effective as in healthy individuals. Overall, experimental results from these two patients could indicate a gradient in severity of pathological language switching, where bilinguals can experience clear impairments in language-restricted lexical retrieval without reaching a point of uncontrolled or pathological language switching.

In reference to the clinical profiles of these two patients, key similarities and differences shed light on the relationship between diagnostic assessments and experimental performance. Based on their WAB scores, both Pt 1 and Pt 5 were classified as moderate in terms of the severity of their language disorder. However, their types of aphasia were polar opposites: Pt 1 demonstrated deficits in auditory comprehension and repetition but fluid speech, compatible with Wernicke’s aphasia, while Pt 5 showed a non-fluent pattern of speech but preserved comprehension and repetition, characteristics of transcortical motor aphasia. Thus, while the WAB’s classifications of aphasia may help to orient diagnosis and treatment strategies in the clinical setting for BWAs, their type of aphasia does not necessarily align with preservation or impairment of BLC; rather, the overall severity and pervasiveness of deficits may better indicate whether their control over their different languages is also impacted.

The subsequent set of data, stemming from the remaining patients without problems in maintaining a language in the single-language condition, can be interpreted in terms of associations and dissociations between deficits in language control mechanisms. Among these patients, two showed parallel impacts in said mechanisms. One patient (Pt 3) produced mixing and switching effects within normal ranges as compared to controls but their poor accuracy suggested a general problem in lexical retrieval; this pattern of performance is compatible with the patient’s diagnosis of moderate anomic aphasia. Interestingly, although Pt 3 committed a large number of omission errors across trials (see Table 5), she did so without any clear deficit in language task engagement or disengagement, as she was able to manage cross-language interference and perform switches during the dual-language condition. These results clearly suggest that word production deficits in bilinguals are to some extent dissociated from the deficits in controlling the two languages [23], as the neural models of bilingualism would also suggest. Although these two systems overlap in a number of brain areas [2,5,66], some are specific to language control and the absence of language control deficits in BWAs could indicate that said areas are spared from brain damage. At the other end of the spectrum, we also saw an example where both reactive and proactive control are affected in the case of Pt 2. As we have shown in previous studies with clinical populations [20,21], both control systems can be affected when patients are asked to perform the switching task with the two languages, despite having some spared abilities of switching in the non-linguistic domain. Here, this generalized impact might be explained by different etiology that the patient had compared to the others; this patient’s language deficits developed following a tumor resection whereas the rest of the patients had post-stroke aphasia. While this is again largely speculative given that we did not have extensive information on the extent of the resection, we might hypothesize that the distinct type of brain damage is the source of such wide-spread pathological performance in both domains of language control.

Between these two extremes of parallel effects, we have three patients who show dissociation between reactive and proactive deficits. Despite it being difficult to interpret what drives patients to have problems in one domain or the other, the results seem to suggest that in some cases these two control mechanisms are not a completely unified system. Pt 6 and 7 had deficits in proactive but not in reactive control, whereas Pt 4 showed a large benefit while naming pictures in the dual-language condition but no significant deviation from controls in reactive control. These types of dissociations are not novel findings, as previous studies have described them in some pathological conditions [46] and in healthy individuals. For instance, Seo and Prat [67] have recently found that these processes may rely on different brain areas, where proactive control is more associated with activation of the dorsolateral prefrontal cortex and premotor cortex and reactive control with activation in the anterior cingulate in bilinguals. Ma, Li, and Guo [17] showed that proactive and reactive control are differentially affected by task conditions (preparation time) and their involvement depends on participants’ language dominance. Furthermore, De Bruin et al. [18,19] have shown that when individuals are free to switch at will in dual-language conditions, they have benefits in global language mixing but then exhibit costs for local switches between the two languages, suggesting a certain degree of dissociation between these two control mechanisms.

According to the DMC framework proposed by Braver [39], proactive control helps individuals to maintain two tasks active in dual condition, similar to the concept of a working memory system. In the context of bilingualism, it has been proposed that proactive control involved in maintaining the activation of the two languages in those contexts in which bilinguals are required to switch [20]. Hence, we might initially interpret increased mixing costs in these two patients (Pt 6 and Pt 7) as a consequence of a deficit in dealing with the activation of two languages, specifically in conflict monitoring as Abutalebi and Green [5] suggest in their framework. However, since the patients did not show any deficit for reactive control (demonstrating similar switch costs to healthy controls), we have to conclude that they had preserved conflict resolution mechanisms as they were able to deal with the cross-language interference in switch trials. Similarly, we can exclude any major difficulties in language engagement or disengagement as they were able to switch from one language to the other at the same rate as healthy controls.

Finally, similar to the two patients with elevated cross-language errors, these five remaining BWAs do not seem to reveal a link between their aphasia types and their performance on the voluntary language switching task. Notably, their aphasia types do not coincide with one another (Pt 2: conduction; Pts 3, 4, and 7: anomic; Pt 6: Wernicke) and have no clear relationship with associations or dissociations in their respective control deficits; this is understandable given that the WAB was not designed to detect deficits in BLC. However, while classification based on aphasia type may not coincide with patterns of BLC deficits, the WAB’s index of severity may, with more severe cases of aphasia corresponding to greater difficulties in language-restricted lexical retrieval. In contrast to the moderate levels of aphasia shared by Pt 1 and Pt 5, four out of five patients in this second group, those patients that did not show deficits in maintaining languages, exhibited only mild forms of aphasia. Observing this emerging pattern, future studies should continue to explore whether severity of aphasia correlates with deficits in maintaining a single language while naming.

In sum, individual patient analyses indicate two profiles of BLC deficits. The first type of BWA here is characterized by decreased capacities in language goal maintenance and/or interference suppression and consequentially committing large numbers of cross-language errors when restricted to one language. While these patients’ deficits in BLC do not reach a level of significance to be classified as “pathological switchers,” they could be viewed as mild cases of pathological language switching that become evident in experimental paradigms. The second, more nebulous type of BWA appears to have preserved functionality in language monitoring but exhibits variability in what aspects of BLC are impacted. Generally, these patients do seem to reveal a certain dissociation between proactive and reactive language control processes.

## 5. Conclusions

The present study aimed at investigating the role of proactive and reactive control in the voluntary language switching through the lens of bilingual aphasia. Results from young individuals replicate previous research while those from patients with aphasia show a more complex picture that required the integration of both group and individual level analyses. Given the complexity of our results, it is difficult to determine whether aphasia affects BLC or not in all patients because this might depend on several factors. However, it is important to highlight two main findings that help both clinicians and future research. First, it is essential to frame the investigation of the BLC deficits in patients with aphasia by using theoretical frameworks of cognitive control [5,39], as they allow us the opportunity to describe spared and affected mechanisms with specificity. Second, a fine-grained analysis of the performance for each patient is essential to identify BLC deficit profiles as well as associations and dissociation between the different language control mechanisms (proactive vs. reactive).

## Figures and Tables

**Figure 1 behavsci-10-00141-f001:**
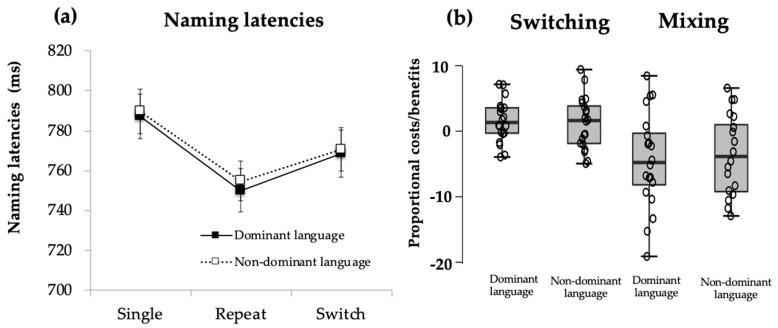
Performance on voluntary switching task for young, healthy Catalan-Spanish bilinguals: (**a**) Naming latencies show significant differences between switch and repeat trials (switch cost) and between repeat and single trials (mixing benefit) with no effect of language. (**b**) Distribution of proportional switching and mixing effects in bilingual participants.

**Figure 2 behavsci-10-00141-f002:**
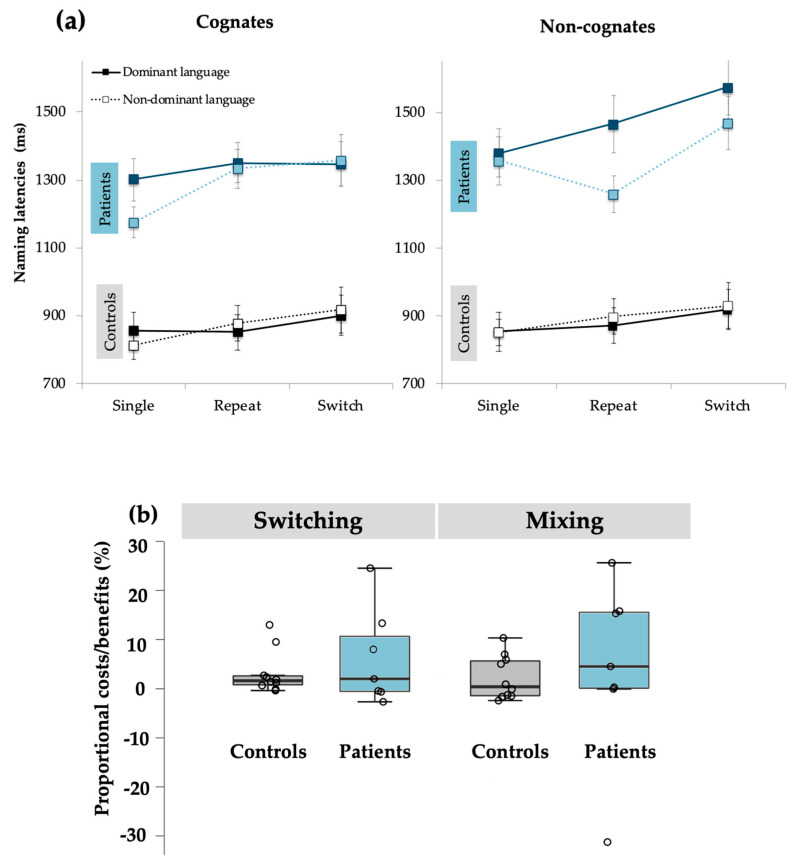
Performance of BWAs and controls on the voluntary language switching task: (**a**) Naming latencies for both experimental groups separated by Trial Type, Language, and Cognate Status. (**b**) Distribution of proportional switching and mixing effects for BWAs and controls.

**Figure 3 behavsci-10-00141-f003:**
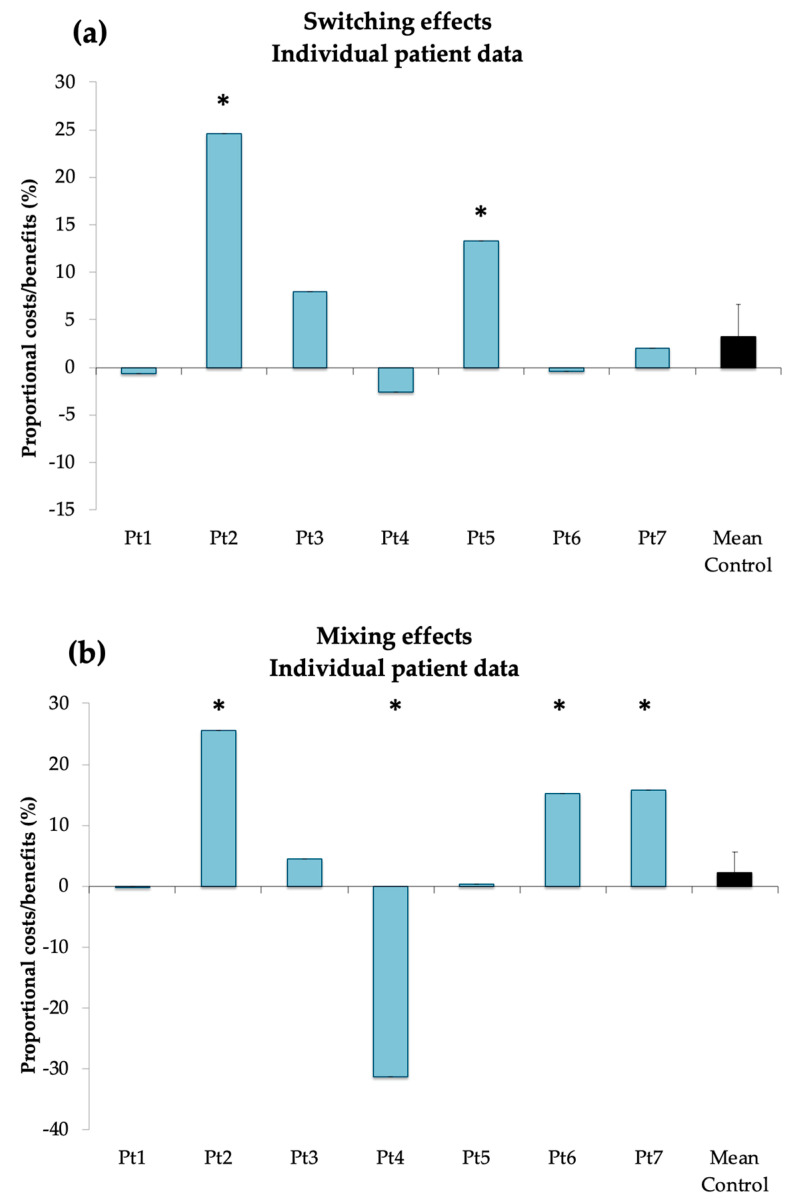
Individual proportional switching (**a**) and mixing (**b**) effects compared to control group mean. Significant deviation from control group mean is marked with an asterisk.

**Table 1 behavsci-10-00141-t001:** Sociodemographic and linguistic measures for young Catalan-Spanish bilinguals.

	M	SD
**Age (years)**	22.35	2.58
**Education (years)**	16.35	2.58
**BSWQ Subscores (max. 15)**		
*DL Switch*	8.65	1.84
*NDL Switch*	8.20	1.91
*Contextual Switch*	8.20	2.61
*Unintended Switch*	6.65	2.74
*BSWQ Overall Switch (max. 60)*	31.70	6.73
**Dominant Language (DL)**		
*Age of Acquisition (years)*	0.75	1.16
*Proficiency (max. 7)*		
*Speaking*	7.00	—
*Comprehension*	6.75	0.64
*Reading*	6.85	0.49
*Writing*	7.00	—
*Language Usage (%)*	54.97	14.48
**Non-dominant Language (NDL)**		
*Age of acquisition (years)*	1.75	1.74
*Proficiency (max. 7)*		
*Speaking*	7.00	—
*Comprehension*	6.75	0.55
*Reading*	6.75	0.55
*Writing*	6.95	0.22
*Language Usage (%)*	34.19	14.10

**Table 2 behavsci-10-00141-t002:** Sociodemographic and linguistic measures for Bilinguals with Aphasia (BWAs) and controls.

	Bilinguals with Aphasia (BWA)		Controls		
	M	SD	M	SD	*p*-Values
**Age (years)**	54.43	6.16	47.67	6.87	0.061
**Education (years)**	14.28	2.93	15	2.16	0.57
**Language Usage (%)**	59.43	10.85	48	20.66	0.203
**BSWQ Subscores (max. 15)**					
*DL Switch*	9.86	1.86	7.8	1.14	0.012
*NDL Switch*	9.57	0.79	6.3	1.57	< 0.001
*Contextual Switch*	9.00	1.41	7.1	2.77	0.118
*Unintended Switch*	7.86	0.69	6.7	2.06	0.176
*Overall Switch (max. 60)*	36.29	3.45	27.9	6.42	0.007
**Dominant Language (DL)**					
*Age of Acquisition (years)*	0	0.00	0	0.00	—
*Proficiency (max. 7)*					
*Speaking*	7	0.00	7	0.00	—
*Comprehension*	7	0.00	6.8	0.42	0.17
*Reading*	7	0.00	6.9	0.32	0.343
*Writing*	6.5	0.85	6.9	0.32	0.191
**Non-dominant Language (NDL)**					
*Age of Acquisition (years)*	4.57	4.08	1.7	2.67	0.098
*Proficiency (max. 7)*					
*Speaking*	7	0.00	7	0.00	—
*Comprehension*	6.75	0.66	6.9	0.32	0.54
*Reading*	7	0.00	6.9	0.32	0.343
*Writing*	6.75	0.66	6.9	0.32	0.54

**Table 3 behavsci-10-00141-t003:** Individual clinical and linguistic data for bilinguals with aphasia (BWAs).

	**Type of Aphasia**	**Months Post-Onset**	**Severity**	**Etiology**	**Dominant Language (DL)**		**Non-Dominant Language (NDL)**	
**Patient 1**	WERNICKE	100	MODERATE	CVA	CAT		SPAN	
**Patient 2**	COND.	156	MILD	TUMOR	CAT		SPAN	
**Patient 3**	ANOMIC	83	MODERATE	CVA	SPAN		CAT	
**Patient 4**	ANOMIC	53	MILD	CVA	CAT		SPAN	
**Patient 5**	TRANS. M.	129	MODERATE	CVA	CAT		SPAN	
**Patient 6**	WERNICKE	114	MILD	CVA	CAT		SPAN	
**Patient 7**	ANOMIC	88	MILD	CVA	CAT		SPAN	
	**Aphasia Quotient (AQ)**	**Spon. Speech**	**Comp.**	**Rep.**	**Naming**	**BAT-C DL (max. = 47)**	**BAT-C NDL (max. = 47)**	***p*-values BAT-C DL vs. NDL**
**Patient 1**	56.3	13	6.25	3.2	5.7	25	15	0.07
**Patient 2**	84.5	18	9.25	6.4	8.6	36	26	0.05
**Patient 3**	71.4	12	7.5	8.1	8.1	35	44	0.02
**Patient 4**	84.1	17	8.75	7.1	9.2	29	31	0.83
**Patient 5**	74.8	12	8.5	8	8.9	28	32	0.51
**Patient 6**	75.7	16	6.75	6.2	8.9	37	35	0.8
**Patient 7**	87.2	15	10	9.7	8.9	41	41	-

**COND.,** Conduction; **TRANS. MOTOR,** Transcortical motor; **CVA,** Cerebrovascular accident; **CAT,** Catalan; **SPAN,** Spanish; **Spon. Speech,** Spontaneous Speech; **Comp.,** Comprehension; **Rep.,** Repetition; **BAT-C,** Bilingual Aphasia Test—Part C.

**Table 4 behavsci-10-00141-t004:** Error distribution for BWA and control groups in single-language blocks. Error types expressed as percentage of overall number of single-language trials.

	Bilinguals with Aphasia (BWAs)	Controls
Omissions	11.07%	3.08%
Cross-language intrusions	5.36%	0.67%
Semantic errors	1.55%	1.75%
Formal errors	1.43%	0.04%
Auto-corrections	1.43%	0.04%
Unrelated errors	0.24%	—

**Table 5 behavsci-10-00141-t005:** Individual error distribution for BWAs in single- and dual-language conditions. Error types expressed as percentage of overall number of condition trials.

**Single-Language**							
	Omissions	Cross-language ^1^	Semantic	Formal	Auto-corrections	Unrelated	**Total**
**Patient 1**	8.33%	21.67%	5.00%	3.33%	—	0.83%	39.17%
**Patient 2**	10.83%	—	—	1.67%	—	—	12.50%
**Patient 3**	25.83%	3.33%	1.67%	—	—	0.83%	31.67%
**Patient 4**	19.17%	0.83%	1.67%	0.83%	0.83%	—	23.33%
**Patient 5**	4.17%	14.17%	—	1.67%	—	—	20.00%
**Patient 6**	4.17%	5.00%	1.67%	1.67%	—	—	12.50%
**Patient 7**	5.00%	1.67%	0.83%	0.83%	—	—	8.33%
**Dual-Language**							
	Omissions	Cross-language ^1^	Semantic	Formal	Auto-corrections	Unrelated	**Total**
**Patient 1**	5.00%	—	6.11%	1.67%	0.56%	—	13.33%
**Patient 2**	7.22%	—	0.56%	0.56%	—	—	8.33%
**Patient 3**	15.56%	—	1.67%	1.67%	—	—	18.89%
**Patient 4**	2.22%	1.11% ^2^	6.67%	1.11%	—	—	11.11%
**Patient 5**	2.22%	—	1.67%	6.11%	1.67%	0.56%	12.22%
**Patient 6**	2.22%	—	1.11%	1.67%	0.56%	—	5.56%
**Patient 7**	7.78%	—	1.11%	2.22%	0.56%	—	11.67%

^1^ Cross-language errors include both cross-language instructions (responding with the correct word but in the incorrect language) and cross-language auto-corrections (committing a cross-language intrusion but correcting with the response in the target language within the allotted time window). ^2^ Patient 4 committed 2 cross-language errors in the dual-language condition by naming the same item correctly twice, but in English.

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
