# Peer review of "Voluntary Language Switching in the Context of Bilingual Aphasia"

_behavsci, 2020, doi:10.3390/bs10090141_

Round 1

Reviewer 1 Report

The present article aims at exploring switching in bilinguals with and without aphasia. The question is addressed by means of two experiments. The first experiment, including participants without neurological damage, replicates a previous study analyzing the performance of bilingual speakers of two typologically different languages and, hence, it provides further evidence of the consistency of these results in speakers of closely related languages. The second experiment explores the performance of people with aphasia.

Although the results are not only interesting from a theoretical perspective but also because of the possible implications for language therapy, the reduced number of informants and more importantly, the huge internal differences in the aphasia group (aphasia type, severity, etiology, language dominance...) makes it problematic for publication in its current form (The authors acknowledge these limitations in the text). Given all differences, the relevance and replicability of groups results is, at least, questionable. A series of case studies instead of a report of group results is highly advised.

In what follows, I add some comments and suggestions(many of them (very) minor) by section.

The title is quite ambiguous with respect to the main aim of the paper and the target population. Does the paper aim at evaluating voluntary language switching tasks, evaluating switch costs and mixing benefits or proactive and reactive control?

Introduction:

L. 56 & 63: consistency in citation form [22] & [23]

L. 58 & ff.: "patient performance on the task was exactly the same as their performance" - singular, it is just one participant.

L. 71: "Language" instead of "speech" disorder

Experiments:

L. 209: delete extra . after )

L. 213-14: "they were presented with experimental pictures"- how many examples? all of them? Otherwise, specify how were stimuli selected - synonyms and dialectal variation may have an effect, a.o. in the cognate non-cognate classification. L. 216: "to promote naming agreement of stimuli across participants" - the selection of unambiguous stimuli via a previous name agreement task would have helped controlling for the possible "semantic errors".

L. 217: By the time participants get to the dual-language block, they have been exposed to the same stimulus many times. This could potentially have a learning effect visible in response times and errors. e.g. Accuracy L. 291 or 299: participants may be showing improvement after successive expositions to the same stimulus (even more redundant for cognates), thus justifying the reduction in number of errors)

L. 340: It is curious to include 3 participants with anomia to perform naming tasks. What motivates this decision? How can naming difficulties be disentangled from factors associated to bilingualism? Do the other participants display similar naming issues?

L. 371: same as line 209

L. 458: "As we observed a great variability in the patient group for switching and mixing effects, we ran individual level analyses for BWAs" - If this is the case, why group level analyses mixing fluent and non-fluent mild and moderate participants with and without anomia? Why not opting for more detailed single case analyses? Such a variability makes it very difficult to “buy” group results, especially considering that 3 out of the 7 participants have anomic aphasia.

L. 484: "controls performed with the same accuracy for cognates (96.20%) and non-cognates (97.40%, p = .37)" - These results contradicts the results reported in experiment 1. (See also L. 562)

L. 494: Are cross-language intrusions more prominent in the NDL?

L. 507: "the second most common errors behind omissions for 2 BWAs (Pt 3 and Pt 7)" - not unexpected given that they have anomic aphasia (see also patient 4 in table 5)

Discussion:

L. 537: "participants switched and repeated languages trial by trial in the dual-language 537 condition at a balanced rate, with switching frequencies remaining around 50% for both languages" - It could be argued that this is actually part of the instruction of the task.

L. 569: the Adaptive Control Hypothesis - given its relevance for the discussion, it should be presented in the introduction

L. 599: "our BWAs have spared functionality in these brain areas" - Adding the lesion location to the profile of the BWAs would make this argument more solid. (see also L. 697)

L. 605: "lexical retrieval deficits" - This is not necessarily the case for the person with a non-fluent aphasia

L. 608: "discard articulatory processes" - The presence of comorbid issues should be added to the profile of the participants (e.g. apraxia, dysarthria...). If not present, this should not be an issue.

L. 646: on instead of to

L. 650: It could be argued that Pt 3, Pt 4 and Pt 7 are anomic and, if they display the same difficulties in both languages (as claimed by the authors), there would be no advantage in using one language over the other (as clarified below).

L. 739: “pathological switchers” - considering that one task forces the participants to switch and that some of them are anomic, checking their spontaneous speech would be a good option to give support to this argument.

All in all, the treatment of BWAs as a group leads to highly inconclusive results and speculative explanations that can be more accurately addressed by presenting the results by case (or subgroups with similar findings) and by providing more background information about the participants (e.g. lesion location).

Reviewer 2 Report

This study aimed to sought to explore voluntary switching in bilinguals with aphasia as well as in healthy bilinguals. The results of study illustrated a complex picture of language control abilities, indicating varying degrees of association and dissociation between factors of BLC. This study is well written overall. The experiment was also properly conducted. If minor revisions are completed, I think it will be a better study.

 1.abstract: It is appropriate to omit the source of the reference in'abstract'. ex.  In Experiment 1, we replicated results of switch costs and mixing benefits reported by de Bruin et al. (2018) within our own bilingual population of Catalan-Spanish bilinguals.

2.The theoretical background and research objectives are well described in the introduction section.

3. The full name of the abbreviation must be indicated in the footnote of the table. ex. CVA

Round 2

Reviewer 1 Report

The authors have addressed all main concerns in their "Response to the reviewer" document.

There are no additional comments with the exception of two very very minor remarks:

l. 214 - 215: "While stimuli with high name agreement were selected for this experiment, this initial presentation of pictures served to strengthen this name  agreement of stimuli across participants." The sentence sounds quite unnatural, reformulation is adviced.

l. 529: bilinguals instead of bilingual